# Formation Behaviors of Coated Reactive Explosively Formed Projectile

**DOI:** 10.3390/ma15248886

**Published:** 2022-12-12

**Authors:** Yuanfeng Zheng, Haiyuan Bie, Shipeng Wang, Peiliang Li, Hongyu Zhang, Chao Ge

**Affiliations:** 1State Key Laboratory of Explosion Science and Technology, Beijing Institute of Technology, Beijing 100081, China; 2Xi’an Institute of Electromechanical Information Technology, Xi’an 710065, China

**Keywords:** shaped charge, liner, formation, reactive materials, coated reactive EFP

## Abstract

The formation behavior of coated reactive explosively formed projectiles (EFP) is studied by the combination of experiments and simulations. The results show that the coated EFP can be obtained by explosively crushing the double-layer liners, and the simulation agrees with the experiment well. Then, the interaction process between the two liners is discussed in detail, and the formation and coating mechanism are revealed. It can be found that there are three phases in the formation process, including the impact, closing and stretching phases. During the impact phase, the velocities of two liners rise in turns with the kinetic energy exchange. In the closing phase, the copper liner is collapsed forward to the axis and completely coats the reactive liner. It is mentioned that the edge of the copper liner begins to form a metal precursor penetrator in this stage. During the stretching phase, the coated reactive EFP is further stretched and fractured, resulting in the separation of the metal precursor penetrator and the following coated reactive projectile. Further studies show both the edge thickness and the curvature radius of the copper liner have significant influences on formation behaviors. By decreasing the edge thickness or the curvature radius, the difficulty of closing decreases, but the tip velocity and the length of precursor penetrator increases. As the thickness and diameter of the reactive liner decrease, the coating velocity increases slightly, but the total length of coated reactive EFP tends to decrease.

## 1. Introduction

The traditional metal EFP (such as copper EFP) mainly relies on a single kinetic energy penetration mechanism to achieve mechanical perforation, but the behind-armor damage effects need to be further improved [1,2]. At the same time, the fluoropolymer-based reactive liner has been studied and developed rapidly in recent years. Its remarkable feature is that the reactive liner is an integral solid energetic liner prepared by mixing energetic powders, cold pressing and sintering [3,4]. Subject to shaped charge effects, the reactive liner is likely to be formed as reactive jet, reactive EFP, or other reactive penetrators. These reactive penetrators can not only perforate targets by its own kinetic energy, but also produce violent behind-armor damage by releasing much chemical energy and gaseous products [5]. However, it should be emphasized that the density and ductility of these reactive penetrators are not sufficient, which results in the penetration depth being dissatisfying. In a word, a copper EFP probably produces deep penetration with relatively weak behind-armor effects, which is just opposite to a reactive EFP. Thus, it is necessary to combine the two types of liners through structural design, to consider both the penetration depth and the behind-armor damage.

Some related studies are as follows: Richard Fong et al., mentioned a composite penetrator of a precursor projectile and a trailing reactive material. Generally, the metal liner with variable thickness is driven by a high-energy explosive to coat the spherical reactive material at the axis to form a penetrator. Then, the metal jet in front uses kinetic energy to penetrate the target, and the following reactive material enters the target and chemical reactions occur, enhancing the damage to the target [6]. Steven Nicolich et al. also introduced the energetic material to enhance the EFP. When the shaped charge is detonated, the energetic material is wrapped inside the copper coat, which would like to improve the damage effects to armored target [7]. When the reactive material is used as the shaped charge liner, the reactive liner is easy to break and reacts in advance under the action of the explosion load. In order to solve this problem, Wan Wenqian [8] and Xin Chunliang [9] et al. installed a buffer layer between the reactive liner and the explosive, which could avoid the reactive liner from being broken directly under the explosion load and reduce the reaction degree of the reactive material in the forming process to a certain extent. Hereafter, Wang Shu-you et al., designed a double-layer coated liner structure called WEFCP (Wrapping Explosively Formed Compound Penetrator). With a small length to diameter ratio, thin edge, and thick center liner structure, the metal liner wraps the reactive material to form WEFCP [10]. Further, the formation process of the WEFCP and terminal effects on the steel target were both studied [11]. Subsequently, Han Yangyang designed a kind of coated liner with variable thickness and carried out numerical simulation. The results show that the coated EFP has lateral effects on the target penetration, and compared with the EFP penetration of the copper liner, the diameter of both the entry hole and the exit hole increases [12]. Using the method of combining experiment and numerical simulation, Su Chenghai carried out the research on the combined damage effect of reactive jet on concrete target penetration and explosion and revealed the influence mechanism of stand-off on the damage effect [13]. Afterwards, Q. Z. Xu et al. designed a wrapping explosively formed penetrator based on a pre-folded double liner, in which the outer liner of higher density forms the shell and the inner liner of lower density forms the core. The following simulations showed that the material strength of double liners is critical for the formation and velocity distribution of wrapping an explosively formed penetrator, which can create a strong lateral effect [14]. Ma H.B. also established a typical reactive material projectile reinforced shaped charge structure. The formation and penetration process of reactive inner core composite EFP was numerically simulated and analyzed [15]. Zhang Xue-peng et al., obtained a type of coated reactive penetrator through simulation studies as well. By adopting a composite charge structure, in which the outer layer is a high-velocity explosive and the inner layer is a low-velocity explosive, the coating effect of the copper liner on the PTFE/Al reactive liner is realized [16].

In summary, previous studies mainly focused on the formation and damage behavior of reactive EFP, including the structure design and damage verification of reactive liners. However, the formation and coating mechanism of the coated reactive EFP have not been studied profoundly. In addition, the influence on the formation behaviors also needs to be further revealed. Therefore, in order to study the forming process of coated reactive EFP and reveal the forming mechanism, this study mainly adopts the method of combining simulation and X-ray experiment [17] to study the above problems, and designs a shaped charge of double-layer liner structure. In this shaped charge structure, the copper liner with varying thickness is used as the outer liner, and the sub-caliber reactive liner is used as the inner liner. In the research process, a pulse X-ray test was used to photograph the formation process, and the corresponding numerical simulation study was carried out. The changes of the velocity and other parameters of the liner were analyzed, and the influence of different liner sizes on the formation of the coated reactive EFP was explored.

## 2. Formation and Coating Behaviors

### 2.1. Shaped Charge with Double-Layered Liners

The coated reactive EFP, depicted in Figure 1, mainly consists of an explosive, a shell, double-layer liners and a baffle ring. The high explosive 8701 is poured into the press mold and a pressure load of 200 MPa is applied at room temperature; thus, the high explosive charge has a length of *l* = 50 mm, a diameter of *d*_1_ = 50 mm, a mass of 74 g, and a density of 1.71 g/cm^3^. The explosive would be centrally initiated through a simple detonator at the bottom. In the experiment, a double-layer liner structure is used, including an outer copper liner and an inner sub-caliber reactive liner. It should be noted that the liner near the explosive is considered as the outer liner and the other end is marked as the inner liner. The copper liner has an outer curvature radius of *R* = 45 mm, an inner curvature radius of about 63.39 mm, a center thickness of *h*_1_ = 3 mm, and an edge thickness of *h*_2_ = 1 mm. The reactive liner has a curvature radius of *R* = 45 mm and a thickness of *h*_3_ = 3 mm. The copper liner and the reactive liner are bonded with shellac varnish. The annular baffle ring, made of 45# steel and used to prompt the detonation products flowing radially, is *a* = 2 mm thick and high. The shell is made of nylon and has a thickness of *s* = 2 mm. The typical photographs of coated reactive EFP is shown in Figure 2. The schematic diagram of the experimental setup is shown in Figure 3a, and the corresponding photograph is shown in Figure 3b. As can be seen, the coated reactive EFP is positioned on a long and hollow cylinder, and an X-ray system is used to capture the formation of coated reactive EFP at times of *t*_1_ and *t*_2_, where the time *t* = 0 corresponds to the moment of the detonator being initiated. The protective plate is also used to prevent the coated reactive EFP from impacting the cement floor.

The preparation process of the reactive liner used in the experiment mainly consists of 3 steps: powder mixing, molding and sintering hardening. At first, the 65.8% PTFE powder, 10.5% copper powder and 23.7% aluminum powder were put into the planetary centrifugal mill for 3 h at room temperature to ensure that the powder of each material was mixed evenly. Then, 3.36 g of reactive material powder were weighed and put into the mold. The static molding is performed through the press, loaded at a velocity of 5 mm/min with a maximum loading pressure of 200 MPa and a holding time of 30 s. Subsequently, the PTFE/Al/Cu reactive liner samples obtained by cold pressing molding were sintered in a vacuum furnace. Firstly, the heating-up stage was carried out from room temperature to 380 °C in about 7 h, at which time the furnace temperature was higher than the melting point of the PTFE matrix and the PTFE matrix melted, keeping this temperature constant for 2 h to make the PTFE matrix fully melted. After that, the furnace temperature dropped to 300 °C within 1 h and continued to keep warm for 5 h, and the PTFE matrix was gradually solidified. Finally, the PTFE matrix recrystallized as the furnace cooled.

### 2.2. Simulation Method

Reactive EFP is modelled and the SPH algorithm is employed in the AUTODYN-3D simulations, and in order to ensure the calculation accuracy and efficiency, a quarter model is established. The size of SPH particles is 0.5 mm, the particles number of the shell, explosive, copper liner, reactive liner and baffle ring are, respectively 26,942; 87,349; 6354; 2662 and 1388, and the initiation point is set at the bottom of the charge. A typical model is illustrated in Figure 4.

The 8701 explosive is selected as the explosive, and the JWL state equation is used to describe the detonation of the charge and the expansion process of the detonation products. The detonation pressure *P* of the explosive is expressed as [18]:(1)$$P=A(1-\frac{\omega }{R_{1}V})e^{-R_{1}V}+B(1-\frac{\omega }{R_{2}V})e^{-R_{2}V}+\frac{\omega E_{0}}{V}$$
where *A*, *B*, *R*_1_, *R*_2_ and *ω* are material constants, *E*_0_ represents the detonation energy per unit volume and *V* is the relative volume. The corresponding parameters of 8701 explosive are from Ref. [18], in which *ρ*_0_ = 1.71 g/cm^3^, *A* = 524.23 GPa, *B* = 7.678 GPa, *R*_1_ = 4.2, *R*_2_ = 1.1, *ω* = 0.34, *E*_0_ = 8.499 GPa, CJ detonation pressure *P*_CJ_ = 28.6 GPa, and detonation velocity *D* = 8315 m/s.

The SHOCK equation of state is used to describe the behavior of materials, including copper, 45# steel and nylon, while the formation process of the coated reactive EFP does not involve the explosive reaction of the reactive material, so the SHOCK equation of state is used to describe the formation behavior of the reactive material.

The Johnson-Cook material model is used to represent the strength behavior of materials, typically metals, subjected to large strains, high strain rates and high temperatures. Such behavior might arise in problems of intense impulsive loading due to high velocity impact. With this model, the yield stress varies depending on strain, strain rate and temperature. The model defines the yield stress *σ* as [19]:(2)$$\sigma =\left(A^{\prime }+B^{\prime }\varepsilon_{p}^{n}\right) [1+C\ln\left(\frac{{\dot{\varepsilon }}_{p}}{{\dot{\varepsilon }}_{0}}\right)] [1-{\left(\frac{T-T_{room}}{T_{melt}-T_{room}}\right)}^{m}]$$
where *A*′, *B*′, *C*, *n* and *m* are material constants. $\dot{\varepsilon }$*_p_* is the effective plastic strain. $\dot{\varepsilon }$_0_ = 1 s^−1^ is the reference plastic strain rate. *T_melt_* and *T_room_* denote the melting and room temperatures, respectively. The detailed parameters for copper and 45# steel are listed in Table 1 [19]. The von Mises yield criterion are used to describe the dynamic response of the nylon shell. Additionally, the shear modulus of nylon G is 3.68 GPa, and the yield stress Y_0_ is 50 MPa.

The main material parameters of the reactive liner are derived and listed in Table 2, based on the Voigt-Reuss-Hill mixing rules [20]. The parameters *ρ*_ini_ is the initial density, *c* and *s* are Hugoniot constants, *Γ* is the Mie-Grüneisen coefficient, *c*_v_ is the specific heat, *G* is the shear modulus, and *Y*_0_ is the yield stress.

### 2.3. Comparison between Experimental and Simulated Results

A comparison between the experimental and simulated results is presented in Table 3. The exposure times of X-ray are 30.8 ms and 50.7 ms. Experiments show that the shape of the coated reactive EFP simulated by the SPH method is similar to the structure of the penetrator in the pulsed X-ray photo. It can be seen from the numerical simulation that the copper liner can completely coat the reactive liner to form a jet-like precursor penetrator and a following coated projectile. The reactive material is mainly distributed in the tail of the coated reactive EFP, forming a shape with a small precursor diameter and a large tail diameter, which are gradually elongated during the overall movement.

At the same time, the size and velocity obtained by numerical simulation are compared with the actual data measured by the X-ray test, and the error is about 10% or less, which shows the effectiveness of the numerical simulation. It should be remembered that the photo corresponding to 30.8 ms does not completely record the trailing form of the coated reactive EFP, which causes the loss of individual data, so only its average tip velocity according to this time can be derived in the photo. The morphologies of coated reactive EFP revealed in the X-ray photographs, which are taken at two different instances in the formation process, are essentially identical to those obtained from the numerical simulations, thus confirming the validity of the numerical simulations.

## 3. Formation Mechanism

In order to investigate the formation mechanism of the coated reactive EFP, observation points are set on the contact surface between the outer copper liner and the inner reactive liner, as shown in Figure 5a, the axial and radial velocity curves shown in Figure 5b,c are obtained. The impact phase is illustrated in Figure 6. Considering the impact on the outer copper liner and the inner reactive liner, the center of the outer copper liner is firstly accelerated by the shock wave, which is transmitted to the inner reactive liner and reflected at the free surface to form a tensile wave. However, the edge of the outer copper liner is driven by the following detonation wave and corresponding gaseous products shortly afterwards. Owing to contrast of thick liner center and thin liner edge, the outer copper liner gains a higher axial velocity at the edge than at the center, thus allowing the edge of outer copper liner to close at the axis due to its radial velocity. Meanwhile, under the combined actions of liners impact and shock wave, the inner reactive liner is squashed and accelerated during the period of being wrapped by the outer copper liner. It is noted there is an “impact- separation- catch up” process when the inner reactive liner is impacted to accelerate. On account of combined effects of transmitted shock wave and reflected tensile wave, the inner reactive liner gains a higher axial velocity that is higher than the outer copper liner; subsequently, a gap is generated between the outer copper liner and inner reactive liner at the axis. The outer copper liner continues to accelerate owing to the actions of the detonation products and quickly catches up to the inner reactive liner, which causes a second impact and then, a second separation. Therefore, the “impact- separation- catch up” process will carry on until the detonation products have no significant effect on the outer copper liner, accompanied by an exchange of kinetic energy, as shown in Figure 5b. It should be noted that the detonation time of explosive is set as “time 0”, which is also expressed as “ *t* = 0”.

Then, the formation of coated reactive EFP enters the closing phase, which is illustrated in Figure 7. Combined with Figure 5, it can be seen that the axial velocity difference of the outer copper liner and the inner reactive liner reduces, meanwhile, the radial velocity difference still exists. Therefore, both the copper liner and the reactive liner are closing continuously, and the closing velocity decreases from the edge to the center of the axis. Thus, the tip micro-element composed of the copper liner edge is closed first, followed by the rest of the copper liner and the reactive liner. After the liners are closed, part of the radial kinetic energy of the liner cell is used to overcome the yield strength of the material and converted into the plastic deformation energy, and the rest of the radial kinetic energy is converted into the axial kinetic energy. At this stage, its shape has changed greatly, and the formation process has been basically completed.

In order to study the internal pressure and temperature changes of the reactive liner during the impact phase and the closing phase, typical observation points are set at the outer and inner side of the reactive liner, as shown in Figure 8a,b, and the pressure curve of the corresponding point from 0 to 40 μs was obtained as shown in Figure 8c; the temperature curve from 0 to 40 μs is shown in Figure 8d. It can be seen from the figure that the pressure inside the reactive liner during the formation process can reach 3 GPa, which mainly occurs when the copper liner strongly impacts the reactive liner, but the peak pressure has a very short duration of action. Additionally, it can be seen that the high pressure area is concentrated on the outer surface of the reactive liner near the copper liner. Meanwhile, the temperature jump also occurs in the impact phase, when the temperature of the outer surface of the reactive liner also jumps greatly, up to 600 K. As the impact proceeds, the internal temperature of the reactive material rises and falls by a small degree, and at the end of the impact phase, the temperature drops slightly and tends to stabilize.

Subsequently, the end of the closing phase is marked by that the radial velocity of the copper liner and the reactive liner both approaching zero, and it can also be seen from Figure 9 that the diameter size of the coated reactive EFP no longer changes significantly thereafter. Then, the coated reactive EFP enters the stretching phase, and there is a velocity gradient along the axial direction of the coated reactive EFP. For example, in the early stage of the stretching phase, the velocity difference between the tip and the tail can reach 800 m/s. Finally, due to the velocity difference between tip and tail, the coated reactive EFP elongates continuously and fractures when the tip copper metal reaches the metal tensile stress limit, forming a metal precursor penetrator and a following coated reactive projectile.

Above all, the formation phases of the coated reactive EFP can be mainly divided into the impact, closing and the stretching phase. The velocity of the copper liner edge is higher than the velocity at the center; ultimately, the reactive liner is wrapped to form a metal precursor penetrator and following coated reactive projectile.

## 4. Influence Mechanism

As the inner liner and the outer liner, the shape parameters of the copper liner and the reactive liner have an important influence on the coating effect. In the following section, the influence of the size parameters of the two liners is investigated.

### 4.1. Edge Thickness of the Copper Liner

The thickness distribution at the center and edge of the copper liner affects the velocity distribution at the corresponding position of the charge liner when the explosive is detonated. In order to study the influence characteristic under different edge thicknesses, holding the center thickness of the copper liner constant, change the edge thickness to 0.7 mm, 1.0 mm, 1.3 mm, 1.6 mm and 1.9 mm, respectively. It can be found that the tip closing velocity of the coated reactive EFP decreases with the increase of the edge thickness, and the time for the tip to complete the closing behavior is also positively correlated with the increase of the edge thickness.

Considering the whole process of formation, the moment of 80 microseconds *t* = 80 μs when the closing stage is about to end is selected to study the influence of different factors on the shape of coating formation. When *t* = 80 μs, the shapes of the coated reactive EFPs formed by different edge thicknesses of the copper liner is shown in Figure 10. As can be seen from the figure that the edge thickness of the copper liner affects the coating effect and the stretching of the jet-like metal precursor penetrator, so the length of the coated reactive EFP decreases as the edge thickness increases. When the edge thickness of the copper liner is small, the edge collapses toward the center while colliding with the reactive liner for kinetic energy exchange, and the reactive liner also undergoes a squeezing and collapsing behavior from the edge toward the center under the collision effect. The copper liner can completely wrap the reactive liner, and the jet-like metal precursor penetrator is fully stretched. However, the edge thickness is small, which makes it difficult to form a metal precursor penetrator with a large diameter. When the edge thickness of the copper liner is larger, the initial velocity of the copper liner edge is smaller. This leads to a poorer coating effect and a further shortening of the jet-like metal precursor penetrator. In particular, when the edge thickness of the copper liner is 1.9 mm, the excessive thickness of the copper liner edge causes the copper edge to not converge at the center, and the velocity difference between the tip and tail of the incompletely coated projectile formed is small. The tip and tail velocities, tip and tail lengths, metal precursor penetrator length and full length of the coated reactive EFP are shown in Figure 11. As the edge thickness of the copper liner increases, the tip velocity, the full length of the coated reactive EFP and the length of the metal precursor penetrator all decrease.

### 4.2. Curvature Radius of the Edge of the Copper Liner

The curvature radius *r*_1_ of the copper liner near the explosive affects the time difference between the detonation wave and the detonation product acting on the edge and the center of the copper liner. In order to study this factor, *r*_1_ is set to 35 mm, 40 mm, 45 mm, 50 mm and 55 mm, and the edge thickness of the copper liner remains the same. As the curvature radius increases, the closing velocity at the tip of the coated reactive EFP gradually decreases, and the time of being wrapped is also positively related to the size of the curvature radius of the copper liner.

When *t* = 80 μs, the shapes of the coated reactive EFPs formed by the different curvature radius of the copper liner is shown in Figure 12. As the curvature radius *r*_1_ of the copper liner increases, the shape of the coated reactive EFP becomes shorter and thicker, and the tip velocity decreases while the tail velocity increases accordingly. When the curvature radius is small, the edge and the center of the copper liner are subjected to a longer interval of impact, and the edge gets a longer acceleration time, resulting in a larger variation in the shape of the copper liner. This way, the coated reactive EFP has a relatively large length-to-diameter ratio and has a long metal precursor penetrator. When the curvature radius is larger, it is obvious that the length-to-diameter ratio of the coated reactive EFP decreases and the shape becomes shorter and thicker. The length of the metal precursor penetrator also becomes significantly shorter. It is noteworthy that when the curvature radius is 55 mm, the coated reactive EFP splits into three segments, which is because the larger curvature radius makes the process of being impacted and impacting of the copper liner more complicated and cannot form the coated reactive EFP with better morphology. The tip and tail velocities, tip and tail lengths, metal precursor penetrator length and full length of the coated reactive EFP are shown in Figure 13. As the curvature radius of the copper liner increases, the tip velocity, the total length of the coated reactive EFP and the length of the metal precursor penetrator decrease.

### 4.3. Thickness of the Reactive Liner

The thickness of the reactive liner is directly proportional to the mass, and the difficulty of coating formation increases as the thickness increases. In order to study this factor, keeping the shaped charge and the copper liner unchanged, the thicknesses of the reactive liner are set to 2 m, 2.5 mm, 3 mm, 3.5 mm and 4 mm, respectively.

When *t* = 80 μs, the shapes of the coated reactive EFPs formed by different thickness of the reactive liner is shown in Figure 14. As the thickness of the reactive liner increases, the mass of the reactive liner also increases, while the tip velocity and the tail velocity gradually decrease. From the figure, it can be found that with the increase of the reactive liner thickness, the full length of the coated reactive EFP grows significantly and the tail velocity decreases significantly. This is due to the fact that as the thickness increases, the mass of the reactive liner increases significantly and its inertia also increases. Hence, the velocity of the copper liner center becomes smaller when impacting the reactive liner with its kinetic energy exchange, resulting in a larger tip-to-tail velocity difference of the coated reactive EFP, which makes a larger elongation of the metal precursor penetrator. It should be noted that when the thickness is 4 mm, the volume of reactive material is large, and it is difficult for the copper liner to cover it completely. The copper liner material is mostly concentrated in the metal precursor penetrator, and the reactive material is exposed in the following coated projectile, which indicates that too large thickness of reactive liner also affects the overall coating effect. The tip and tail velocities, tip and tail lengths, metal precursor penetrator length and full length of the coated reactive EFP are shown in Figure 15. It can also be found that the velocity distribution of the coated reactive EFP is not affected much when the reactive liner thickness is varied within a reasonable range.

### 4.4. Diameter of the Reactive Liner

The diameter of the inner reactive liner also determines the mass and shape of it. In order to study this factor, keeping the shape of the shaped charge and the copper liner unchanged, the radiuses of the reactive liners are set to 20 mm, 22 mm, 24 mm, 26 mm and 28 mm, respectively. Additionally, the corresponding edge closing time of the copper liner increases as the diameter of the reactive liner increases. This is because as the diameter and mass of the reactive liner increases, more kinetic energy is transferred to the reactive liner in the form of collisions under the same explosive driving force, resulting in a decrease in the closing velocity and an increase in the closing time of the copper liner.

When *t* = 80 μs, the shapes of the coated reactive EFPs formed by different diameter of the reactive liner is shown in Figure 16. The tip and tail velocities, tip and tail lengths, metal precursor penetrator length and full length of the coated reactive EFP are shown in Figure 17. As the diameter of the reactive liner increases, the mass of the reactive liner increases and the diameter of the following coated projectile increases accordingly. It is noteworthy that the metal precursor becomes slenderer, which is due to the fact that more copper material is used for coating to form the following coated reactive projectile with a larger diameter reactive liner, and therefore less copper material becomes closed in the center of the axis in advance. It is worth mentioning that when the diameter of the reactive liner is 32 mm, the copper liner could not crush the reactive liner, so the coated reactive EFP broke and the diameter of the following coated projectile is too large.

## 5. Conclusions

Aiming at the structure of the coated shaped charge, the present work studies the formation process and influencing factors of the coated reactive EFP through numerical simulation, and uses the pulse X-ray photography technology to verify the validity of the numerical simulation. The main conclusions are as follows:(1)The formation stage of the coated reactive EFP can be mainly divided into the impact phase, the closing phase and the stretching phase. The edge of the outer variable thickness copper liner is thinner, resulting in the edge velocity higher than the velocity at the axis after the explosive is detonated. The copper liner is closed forward and collides with the sub-caliber reactive liner, thereby forming a metal precursor penetrator with the copper edge as the main body and a large-diameter projectile with coated reactive material.(2)A static explosion experiment is carried out, and the effectiveness of the numerical simulation is verified by combining the X-ray of a typical forming behavior. The experiment showed that the shape of the EFP obtained by simulating by the SPH method is similar to that of the penetrator in the pulsed X-ray photograph.(3)The edge thickness and curvature radius of the variable thickness copper liner, the diameter and thickness of the sub-caliber reactive liner all have an important influence on the formation process of the coated reactive EFP. With the decrease of the edge thickness and curvature radius of the copper liner, the coating velocity of the copper liner and the tip or tail velocity of the coated reactive EFP increase, while the total length of the coated reactive EFP and the length of the metal precursor penetrator increase significantly. As the thickness or diameter of the reactive liner decreases, the difficulty of coating decreases, the tail velocity of the coated reactive EFP increases obviously, and the total length of the coated reactive EFP tends to decrease. If the thickness or diameter is too large, the coated reactive EFP is easy to fracture and not conducive to coating formation.

## Figures and Tables

**Figure 1 materials-15-08886-f001:**
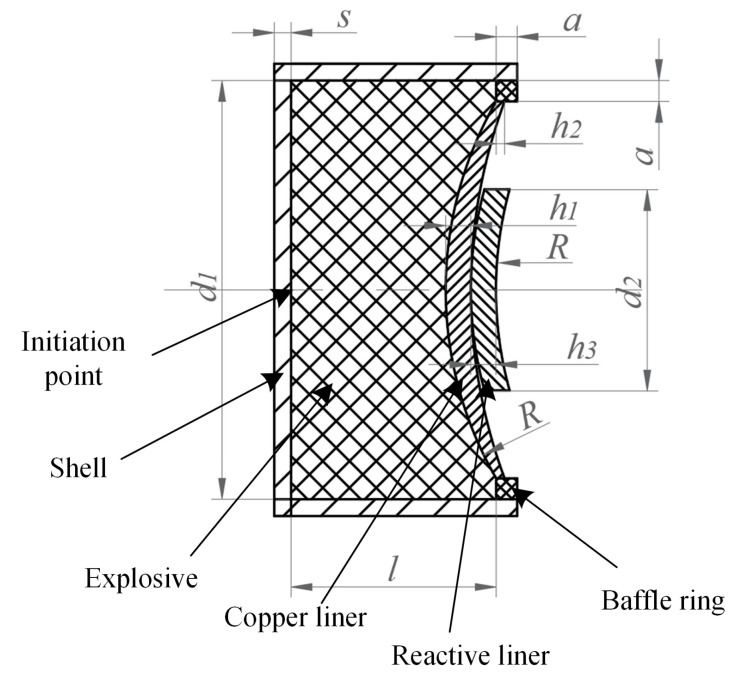
Sketch of coated reactive EFP.

**Figure 2 materials-15-08886-f002:**
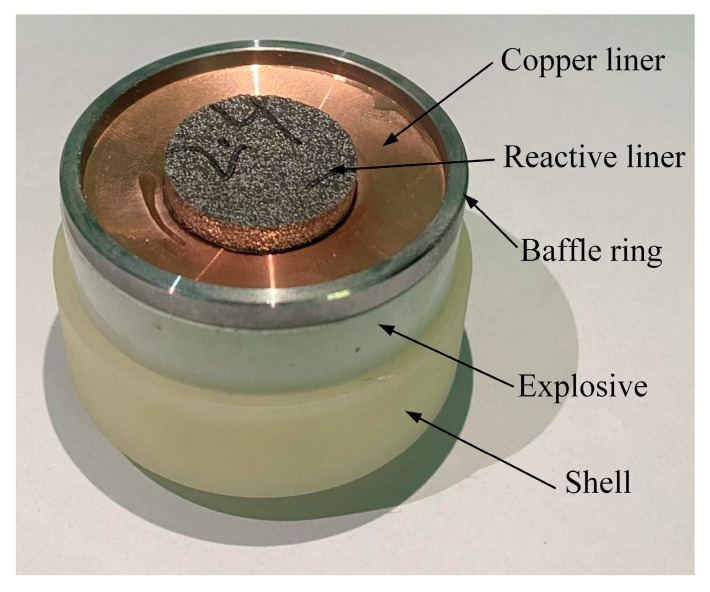
Structure of coated reactive EFP liner.

**Figure 3 materials-15-08886-f003:**
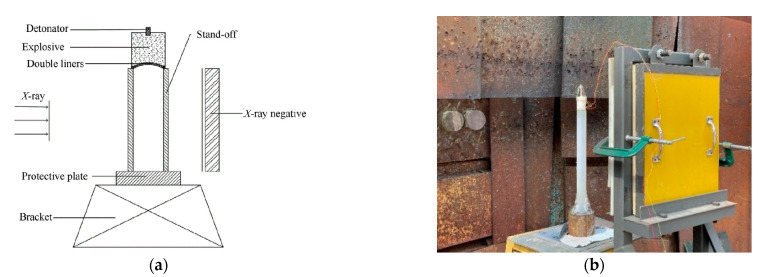
Experimental principle and setup of X-ray: (**a**) principle; (**b**) setup.

**Figure 4 materials-15-08886-f004:**
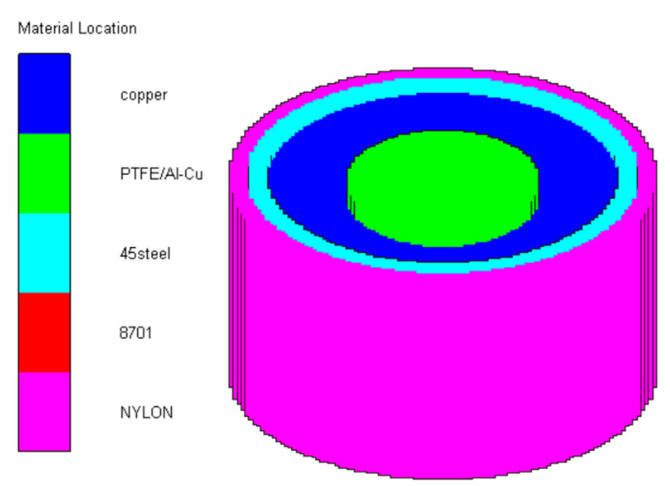
Numerical model of reactive EFP.

**Figure 5 materials-15-08886-f005:**
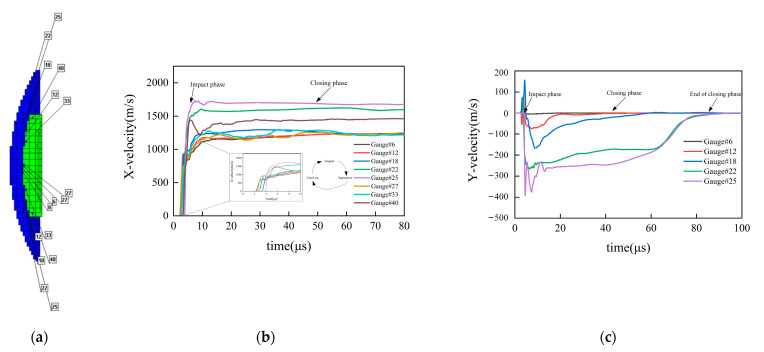
Axial and radial velocity curve of typical observation points: (**a**) Distribution and number of gauge points setting; (**b**) Axial velocity curve of typical observation points; (**c**) Radial velocity curve of typical observation points.

**Figure 6 materials-15-08886-f006:**
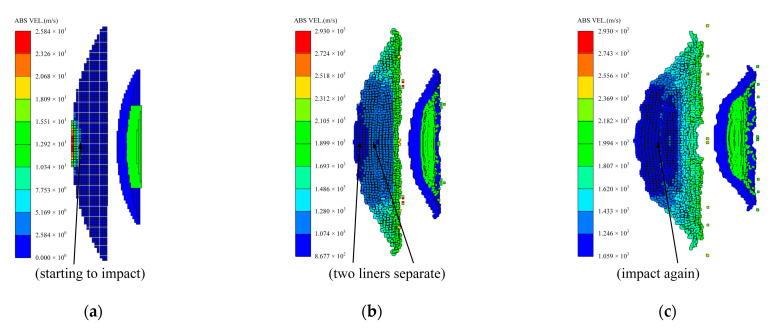
Impact phase: (**a**) *t* = 2 μs; (**b**) *t* = 8 μs; (**c**) *t* = 12 μs; (**d**) *t* = 17 μs; (**e**) *t* = 20 μs.

**Figure 7 materials-15-08886-f007:**
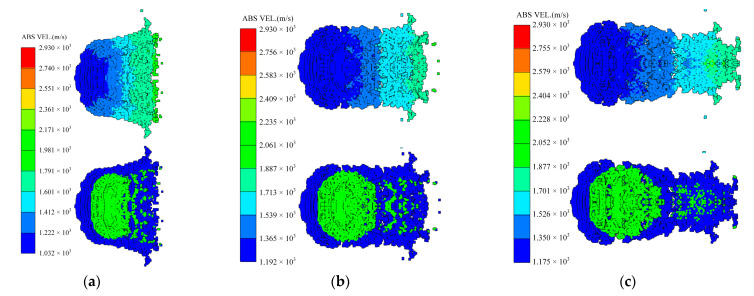
Closing phase: (**a**) *t* = 30 μs; (**b**) *t* = 45 μs; (**c**) *t* = 60 μs; (**d**) *t* = 75 μs; (**e**) *t* = 90 μs.

**Figure 8 materials-15-08886-f008:**
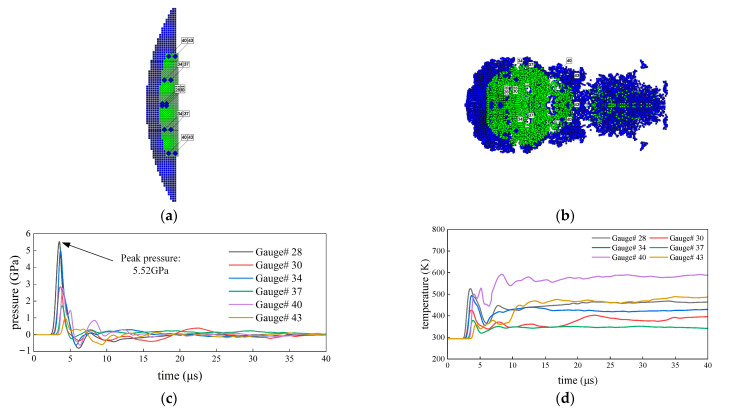
Typical pressure and temperature changes inside the reactive liner: (**a**) Distribution and number of gauge points setting; (**b**) *t* = 60 μs distribution and number of observation points; (**c**) Pressure-time curve; (**d**) Temperature-time curve.

**Figure 9 materials-15-08886-f009:**
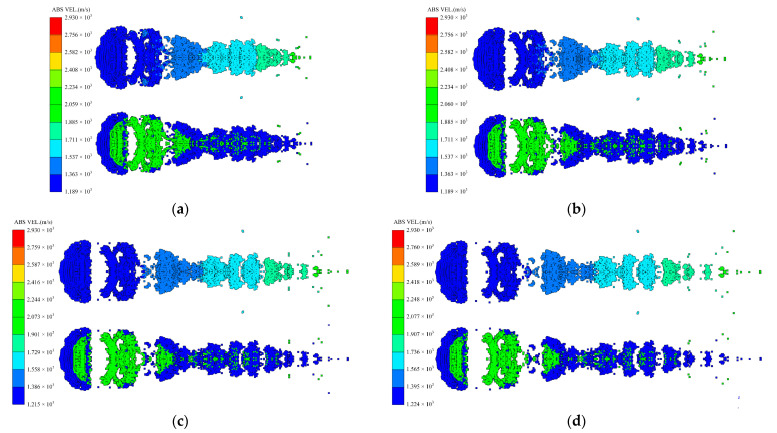
Stretching phase: (**a**) *t* = 100 μs; (**b**) *t* = 115 μs; (**c**) *t* = 130 μs; (**d**) *t* = 145 μs.

**Figure 10 materials-15-08886-f010:**
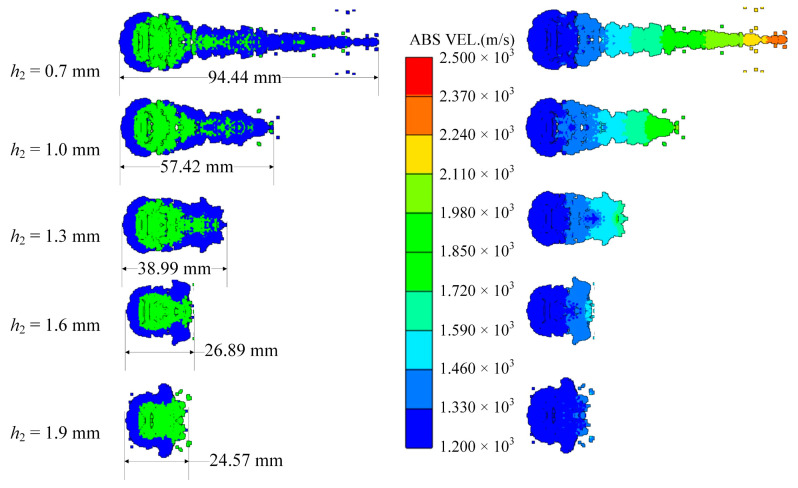
Coating shapes and velocity distribution as the edge thickness of the copper liner changes.

**Figure 11 materials-15-08886-f011:**
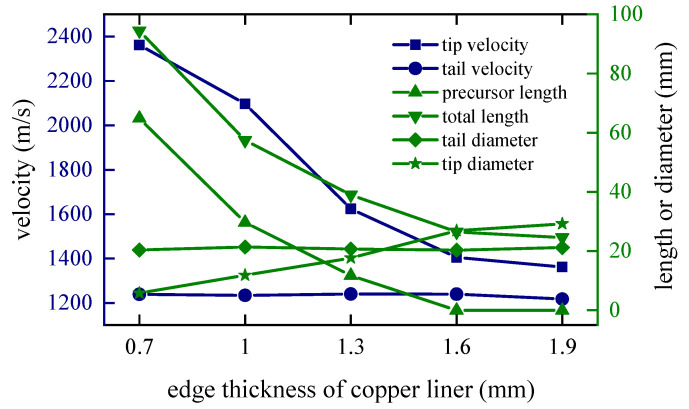
Curves of coating EFP shape parameter change.

**Figure 12 materials-15-08886-f012:**
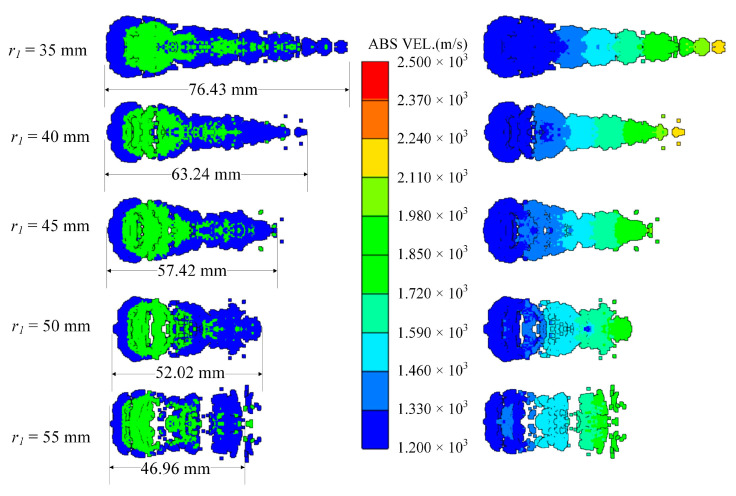
Coating shapes and velocity distribution as the curvature radius of the copper liner changes.

**Figure 13 materials-15-08886-f013:**
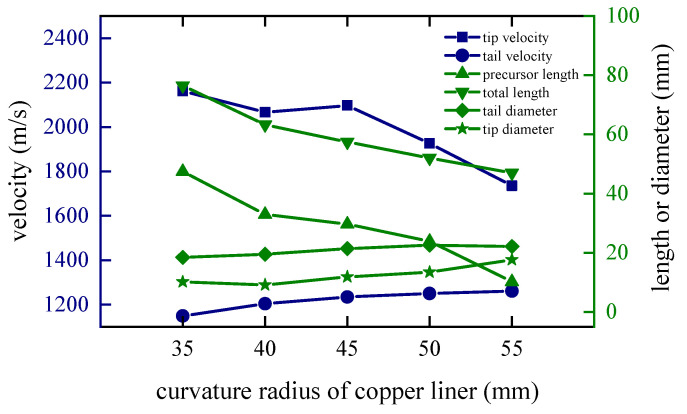
Curves of coating EFP shape parameter change.

**Figure 14 materials-15-08886-f014:**
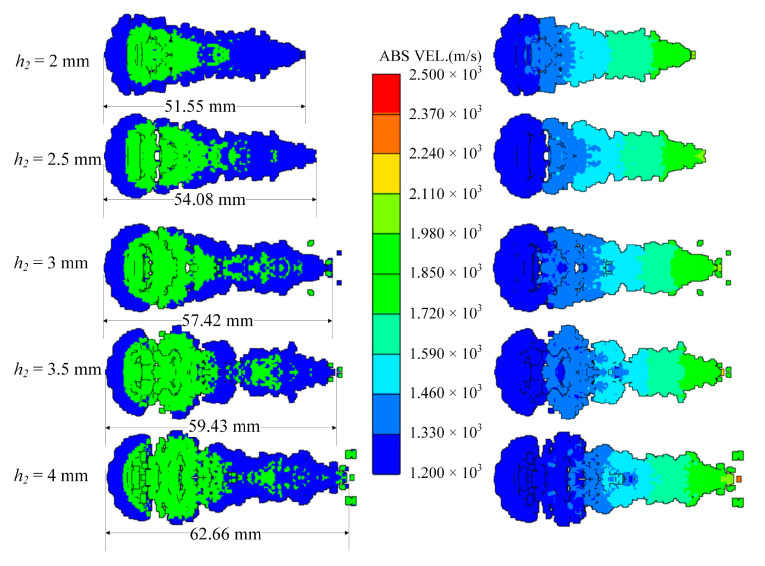
Coating shapes and velocity distribution as the thickness of the reactive liner changes.

**Figure 15 materials-15-08886-f015:**
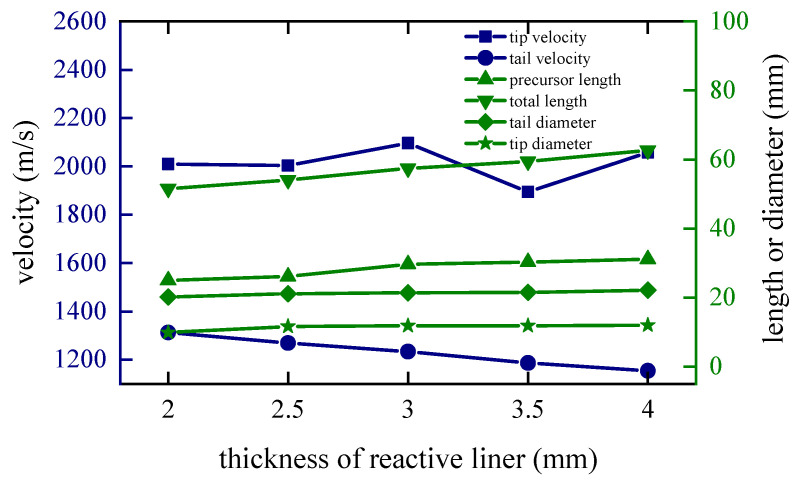
Curves of coating EFP shape parameter change.

**Figure 16 materials-15-08886-f016:**
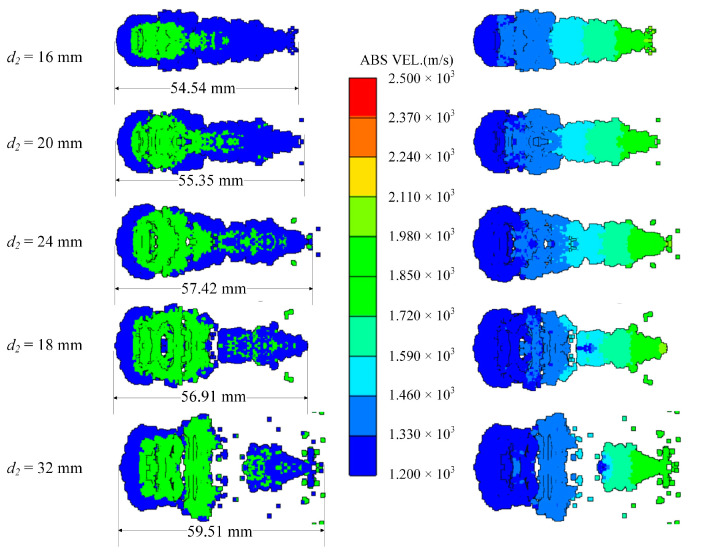
Coating shapes and velocity distribution as the diameter of the reactive liner changes.

**Figure 17 materials-15-08886-f017:**
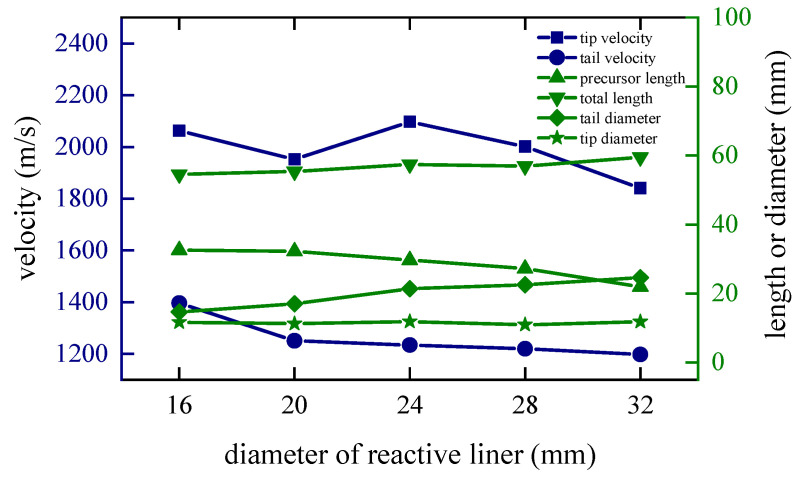
Curves of coating EFP shape parameter change.

**Table 1 materials-15-08886-t001:** Material model parameters.

Material	*Ρ* (g/cm^3^)	*A*′ (GPa)	*B*′ (GPa)	*C*	*n*	*m*	*T* _m_
copper	8.97	0.09	0.292	0.025	0.31	1.09	1356
45# steel	7.89	0.175	0.38	0.060	0.32	0.55	1811

**Table 2 materials-15-08886-t002:** Reactive material model parameters.

Parameters	*ρ*_ini_ (g/cm^3^)	*C* (m/s)	*s*	*Γ*	*c*_v_ (J/KgK)	*G* (MPa)	*Y*_0_ (MPa)
Reactive material	2.46	1711.45	2.18	1.014	1078.4	667	120

**Table 3 materials-15-08886-t003:** Comparison between experiments and simulations.

	*t*_1_ = 30.8 μs		Error/%	*t*_1_ = 50.7 μs		Error/%
	Experiment	Simulation	-	Experiment	Simulation	-
Morphology Comparison	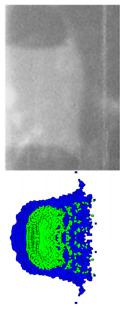		-	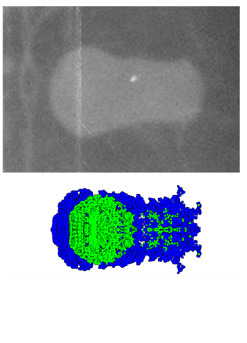		-
Tip diameter (mm)	25.47	26.07	2.36	15.09	16.58	9.37
Tail diameter (mm)	22.17	21.32	−3.83	18.4	20.1	9.24
Total length (mm)	-	-	-	33.49	36.87	10.9
Tip velocity (m/s)	1580.2	1707.8	8.75	1580.2	1710.3	8.23

## Data Availability

Not applicable.

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
