# Peer review of "Formation Behaviors of Coated Reactive Explosively Formed Projectile"

_materials, 2022, doi:10.3390/ma15248886_

Round 1

Reviewer 1 Report

- References should be improved. New references could be added to the manuscript (especially up-to-date papers from the last five years).

- The aim, scope, and novelty of the study should be emphasized in the last paragraph of the Introduction.

- In Table 3, the units of tip diameter, tail diameter, total length, and tip velocity should be indicated.

- In Figure 8, the word "Gauge" can be enlarged a little more for better reading.

- The authors wrote that the Conclusions section is not mandatory. However, according to the Reviewer, this part should not be excluded from the study. On the contrary, this part should be improved.

Reviewer 2 Report

In the present work, the authors study the formation behaviors of coated reactive EFP based on both experiments and numerical simulations. Three phases in process of the formation i.e., impact, closing and stretching phases. Then, the influence of some factors is examined in detail. In general, the manuscript is well written, and the present topic fits into the scope of the journal. The present results are presented as well as discussed in detail. The manuscript can be reconsidered for publication after carefully addressing the following minor issues

First of all, the EFP should be written in explicit form in the title. The title of research papers should not be presented with abbreviated symbols.

Did the authors establish Eqs. (1) and (2) for the first time in this work? If not, please add the appropriate citation.
